# Integrated transcriptome and proteome analysis reveals the unique molecular features and nutritional components on the muscles in Chinese Taihe black-bone silky fowl chicken

Guanghua Xiong[1,2], Wanqing Chen[1], Kai Jiang[1], Shuyuan Liu[1], Juan Li[1], Xinjun Liao[1]*

1 Ji'an Key Laboratory of Genetics, Breeding and Reproduction in Taihe Silky Fowl, College of Life Sciences, Jinggangshan University, Ji'an, Jiangxi, China, 2 Key Laboratory of Embryo Development and Reproductive Regulation of Anhui Province, College of Biological and Food Engineering, Fuyang Normal University, Fuyang, Anhui, China

* xinjun_liao2022@126.com

**Data Availability Statement:** All relevant data are within the manuscript and its Supporting Information files.

## Abstract

The Taihe Black-Bone silky fowl chicken (BB-sfc) is a renowned dietary and medicinal chicken globally recognized for its high nutritional and medicinal value. Compared to the local Black-Bone black-feathered chicken (BB-bfc), the Taihe silky fowl chicken has higher levels of amino acids, trace elements, and unsaturated fatty acids in their muscles, which offer anti-aging, anti-cancer, and immune enhancing benefits. Despite this, the unique nutritional components, genes, and proteins in Taihe silky fowl chicken muscles are largely unknown. Therefore, we performed a comprehensive transcriptome and proteome analysis of muscle development between BB-sfc and BB-bfc chickens using RNA-Seq and TMT-based quantitative proteomics methods. RNA-Seq analysis identified 286 up-regulated genes and 190 down-regulated genes in BB-sfc chickens, with oxidoreductase activity and electron transfer activity enriched in up-regulated genes, and phospholipid homeostasis and cholesterol transporter activity enriched in down-regulated genes. Proteome analysis revealed 186 significantly increased and 287 significantly decreased proteins in Taihe BB-sfc chicken muscles, primarily affecting mitochondrial function and oxidative phosphorylation, crucial for enhancing muscle antioxidant capacity. Integrated transcriptome and proteome analysis identified 6 overlapped up-regulated genes and 8 overlapped down-regulated genes in Taihe silky fowl chicken, related to improved muscle antioxidant status. Taken together, this research provides a comprehensive database of gene expression and protein information in Taihe Black-Bone silky fowl chicken muscles, aiding in fully exploring their unique economic value in the future.

**Funding:** This research was funded by the National Natural Science Foundation of China (Grant No. 82160048), Natural Science Foundation of Anhui Province (Grant No. 2308085MH265), Education Department of Jiangxi Province (Grant No. GJJ2201611), and Science and Technology Research Project of Ji'an City (Grant No. 20211-055455). The funders had no role in study design, data collection and analysis, decision to publish, or preparation of the manuscript.

**Competing interests:** The authors have declared that no competing interests exist.

# 1. Introduction

The development of poultry farming is of great significance for accelerating the construction of modern animal husbandry and improving people's quality of life [1, 2]. The Chinese Taihe Black-Bone silky fowl chicken (BB-sfc) is a famous and national geographical indication product of China with the history of over two thousand years, which originated in Wangbantu village, Taihe County, Jiangxi province [3, 4]. The typical physiological characteristics of BB-sfc is the snow-white feathers, but a large amount of melanin deposits in various tissues including skin, meat and bones [5, 6]. BB-sfc is not only a nutritious food but also has the unique medicinal value, which has been extensively utilized to enhance the immune system, alleviate headaches and hepatitis, as well as cure women's diseases including menstrual incongruity and postpartum complications [7, 8]. Based on the previous studies, it has been found that the breast meat of BB-sfc chicken contains natural melanin and high levels of carnosine, which have shown the potential pharmaceutical function such as free radical scavenging and anti-oxidant effects [9, 10]. However, the poor growth performance and lower egg production ability of BB-sfc restrict the economic benefits of related enterprises compared to the other breeds of chickens.

Since conventional genetic breeding requires long-term selection and generally takes a long time, the modern molecular breeding using the high-throughput sequencing can significantly improve breeding efficiency and shorten the breeding cycle [11, 12]. Transcriptome sequencing (RNA-Seq) is a rapid method for determining the types and abundance of mRNA in cells, tissues, or organisms of interest, which is commonly employed to investigate the differentially expressed genes and regulated signaling pathways [13]. RNA-Seq is extensively utilized to uncover the crucial pathways and identify candidate genes that play a role in poultry meat production and growth performance at the molecular level [14, 15]. Several previous studies have utilized RNA-Seq in poultry research. For example, a study analyzed the ovaries of Taihe silky fowl chicken at three different egg-laying stages using transcriptome sequencing, which identified the differential genes including LPAR3 and SMOC1, as well as important signaling pathways such as proteolysis and vascular smooth muscle contraction through comparative analysis [16]. Another study focused on transcriptome analysis of embryonic muscle development in Chengkou Mountain Chicken, which identified a significant number of differentially expressed genes (DEGs) related to muscle development [17].

Proteomics is a powerful method for studying the composition, distribution, changes, and interactions of proteins in cells, tissues, or organisms [18, 19]. Proteome sequencing mainly relies on liquid chromatography and high-resolution mass spectrometry, which can be used to determine the types and quantities of proteins expressed in cells, as well as the differences in cell populations between different states [20, 21]. Furthermore, quantitative proteomics can identify any type of protein, especially for low abundance proteins with high sensitivity [22]. Mass spectrometry-based comparative proteomics techniques such as Label-free, iTRAQ and TMT, have facilitated the extensive identification of proteins with subtle changes in expression levels [23, 24]. For instance, a study conducted quantitative proteome analysis on chicken breast meat from NEAUHLF chicken lines, revealing the identification of 199 differentially abundant proteins (DAPs) in the lean line compared to the fat line [25]. There have been limited studies on chicken using proteomics research, even though it has been effectively utilized to identify biomarkers and explore molecular pathways associated with meat quality in other animals like pigs, cows, and lambs [26, 27]. By combining proteomics with other omics approaches, more comprehensive data can be obtained to investigate the mechanisms of chicken muscle growth.

Recent advancements in high-throughput sequencing technology have enabled the investigation the biological basis of agriculturally complex traits in farm animals by integrating transcriptome and proteome technologies. In the present study, we explored the variations of transcriptome and quantitative proteome profiling of chicken breast meat between BB-sfc and BB-bfc chickens. We successfully identified multiple differentially expressed genes and proteins, as well as key signaling pathways involved in chicken muscle development. In conclusion, our study offers valuable insights into the differences in muscle composition between Taihe silky fowl chicken lines due to divergent selection.

## 2. Materials and methods

### 2.1. Animal housing and sample collection

The approved methods for chicken raising, handling and collecting samples were authorized by the Ethics Committee of the Institutional Animal Care and Use Committee of Jinggangshan University (Permit Number: ECIACUC-JGSU-20220310). The protocol described the details on (1) methods of sacrifice, (2) methods of anesthesia and/or analgesia, and (3) efforts to alleviate suffering.

The Chinese Taihe Black-Bone silky fowl chicken (BB-sfc) was purchased from Aoxin black-bone silky fowl Development Co., Ltd. (Taihe county, Jiangxi province, China). Meanwhile, the Black-Bone and black-feathered chicken (BB-bfc) were obtained from Ji'an local farmer. The chickens were introduced into the laboratory animal room and reared in the independent cages (78 cm L x 50 cm W x 60 cm H) under the photoperiod cycle of 16 h Light /8h Dark, which free access to the diet and water, respectively. All the chickens were selected and kept under the standard feeding conditions and the normal chicken diet mainly contains 60% corn, 20% soybean meal and 10% wheat bran. In total, eight female chickens (4 BB-sfc and 4 BB-bfc) at 12 weeks of age were euthanized by using a firm percussive blow to the head followed by manual or mechanical cervical dislocation. The chicken breast meat were collected and immediately stored for subsequent analysis.

### 2.2. Library construction and high-throughput transcriptome sequencing

'Total RNA from chest muscle tissues of two breed chickens was extracted using the TRIzol reagent. The RNA quality was detected using NanoDrop® spectrophotometer and Agilent 2100 Bioanalyzer system. The RNA quality in each sample was measured and listed in S4 Table. After that, the cDNA sequencing libraries were prepared as follows: Enrichment of eukaryote mRNA was achieved by utilizing Oligo (dT) magnetic beads from the entire RNA. Following this, the obtained mRNA underwent fragmentation and reverse transcription to generate cDNA. The cDNA was further subjected to end repair, addition of A-tail, and finally connected to the sequencing adaptors. Next, fragment size selection was performed using AMPure XP beads, followed by PCR enrichment to obtain the ultimate cDNA library. Finally, three replicates of BB-sfc and three replicates of BB-bfc cDNA libraries for chicken muscle samples were constructed and raw paired-end reads were generated using the Illumina NovaSeq 6000 sequencing platform from Appiled Protein Biotechnology Co., Ltd. (Shanghai, China).

### 2.3. Bioinformatics analysis of transcriptome data

The paired-end clean reads from all samples were mapped onto the *Gallus_gallus* genome using the Hisat2 software. The mapped reads of each sample were assembled and quantified by StringTie. The abundance of gene expression level is represented by utilizing the featureCounts software to calculate the FPKM value of each gene. Meanwhile, principal component

analysis (PCA) is used to evaluate inter-group differences and the biological repeatability of samples within the group by using eigen package. The differential genes were identified using DESeq2 package. Genes with a corrected p-value < 0.05 and |log$_2$(fold change)| > 1 were considered significant.

The hierarchical clustering analysis of DEGs was performed using the gplots package in R environment. The DEGs were selected and constructed the regulatory enrichment network by using the STRING: functional protein association networks (https://cn.string-db.org/), which was visually represented by using the Cytoscape software. The GO enrichment analysis was performed by the DAVID online tool (https://david.ncifcrf.gov/). The significantly enriched GO terms were identified with p-value was less than 0.05. Besides, KEGG pathway was enriched by using the KOBAS 3.0 (http://kobas.cbi.pku.edu.cn/).

## 2.4. Protein extraction and peptide enzymatic hydrolysis

The complete protein was extracted using SDT (4% (w/v) SDS, 100mM Tris/HCl pH 7.6, 0.1M DTT) lysis method, and then protein quantification was performed using BCA method. Each sample contains an appropriate amount of protein.Trypsin enzymatic hydrolysis was performed using the Filter aid protein preparation (FASP) method [6], and peptide segments were analyzed using C18 Cartridge Perform desalination, freeze dry peptide segments, and add 40 µ Resolution with 0.1% formic acid solution, peptide quantification (OD280). Take 100µg samples for each sample peptide are labeled according to the instructions of Thermo's TMT labeling kit. Each group contains 4 biological duplicate samples, totaling 12 samples.

## 2.5. LC-MS/MS data collection

The experimental setup involved the utilization of an HPLC liquid phase system (Thermo Easy nLC 1200), which enabled the separation of proteins in each sample. The separation involved the use of Buffer A, which was a 0.1% aqueous solution of formic acid, and Solution B, consisting of a 0.1% aqueous solution of formic acid with acetonitrile (comprising 84% acetonitrile). The chromatographic column utilized was the Thermo Scientific Acclaim PepMap100, 100 µM * 2 cm, equipped with the nanoViper C18 technology, and it was initially conditioned with 95% liquid A. Subsequently, the sample was loaded onto the auto sampler for further analysis. The column utilized for chromatographic separation was the Thermo Scientific EASY column, which had a flow rate of 300 nL/min. Following the separation process, the sample was subjected to mass spectrometry analysis using the Q-Exactive mass spectrometer. In this analysis, the scanning range of the mother ion was set at 300–1800 m/z, and a primary mass spectrometry resolution of 70000 at 200 m/z was achieved. The detection technique used is positive ion.

## 2.6. Protein identification and functional analysis

Firstly, normalize the quantitative information of the target protein set. Then, use the Complexheap R package (R Version 3.4) to classify the two dimensions of sample and protein expression, and generate a hierarchical clustering heatmap. Adopting CELLO (http://cello.life.nctu.edu.tw/) method for predicting subcellular localization. The annotation and functional analysis of the target proteins were performed by using Blast2GO online tool. Based on IntAct or STRING, The information in the database is searched for direct and indirect interaction relationships between target proteins, and the interaction network is generated using Cytoscape software (version 3.2.1) for network analysis.

### 2.7. Statistical analysis

The p-value of DEGs, DAPs, GO term and KEGG pathway enrichment were adjusted by the Benjamini-Hochberg (BH) method. The statistical analysis of this study was all carried out with the SPSS 20.0 and comparison between two groups using student t-test. Each experiment had three biological replicates or more and was reported as mean ± SD. For statistical significance, $*$ $p < 0.05$ and $**$ $p < 0.01$ were used.

## 3. Results

### 3.1 Differences in growth performance and multi-omics analysis of Chinese Taihe Black-bone silky fowl

To investigate the potential economic value and unique nutritional components in Chinese black-bone chickens, we used two chicken breeds (BB-sfc and BB-bfc) in this study. Firstly, the most obvious morphological characteristics of BB-sfc is the body covered with white filamentous fluff, tassel head, compound crown, and green ears compared with the BB-bfc chickens (Fig 1A). Secondly, the eggs of BB-sfc are smaller in size and lighter in weight than BB-bfc chickens (36.2±3.1 g vs. 39.5±4.3 g) (Figs 1B and S1). Thirdly, the color of muscles, skin, and bones is darker than black-feathered chickens (Fig 1C). Besides, the two breeds of chickens between BB-sfc and BB-bfc also have significant differences in other growth indicators such as body weight, feed intake, egg production and feed conversion ratio (S1 Table). Taken together, the results suggested that the body appearance, egg weight and muscle color of Taihe black-bone chickens are different from the ordinary black chickens.

To enhance our understanding of the variances in gene expression and protein composition within Taihe black-bone silky fowl, we executed a TMT-based proteomic quantitative analysis with four muscle samples as biological replicates. Additionally, we performed RNA-Seq sequencing analysis on three separate muscle samples. By conducting a comprehensive analysis of the proteome and transcriptome, we successfully identified the distinct genes and proteins present in Taihe black-bone silky fowl, as well as the exclusive genes and proteins specifically found in BB-sfc chicken (Fig 1D). In summary, these identified differential genes have the potential to be targeted for the development of functional specific foods for Taihe black-bone chickens.

### 3.2 Transcriptome analysis of Taihe silky fowl in muscle sample

Overall, a total of 375,325,766 pristine reads were acquired from the six RNA libraries, averaging 62,554,294 unblemished reads per sample (the read count ranged from 53,082,682 to 73,930,666). Furthermore, all the reads were successfully aligned with the reference genome of chickens (Gallus gallus), surpassing an 80% mapping rate (Table 1). Next, the expression level of each gene (FPKM value) was calculated, and the significantly differentially expressed genes (DEGs) between BB-sfc and BB-bfc were identified. The results suggested that a total of 476 DEGs including 286 up-regulated and 190 down-regulated genes were discovered in BB-sfc compared with the BB-bfc group (Fig 2A and S2 Table). Furthermore, the hierarchical clustering analysis indicated that the expression trends of the three replicates of BB-sfc and BB-bfc samples in DEGs were basically consistent, which further proving that our sequencing data had good biological repeatability (Fig 2B).

Principal Component Analysis (PCA) is also commonly used to evaluate inter-group differences and the biological reproducibility of samples within the group. Our results showed that BB-sfc and BB-bfc were clustered in different quadrants in two-dimensional PCA analysis (Fig 2C). In order to further investigate the connections between these differentially expressed

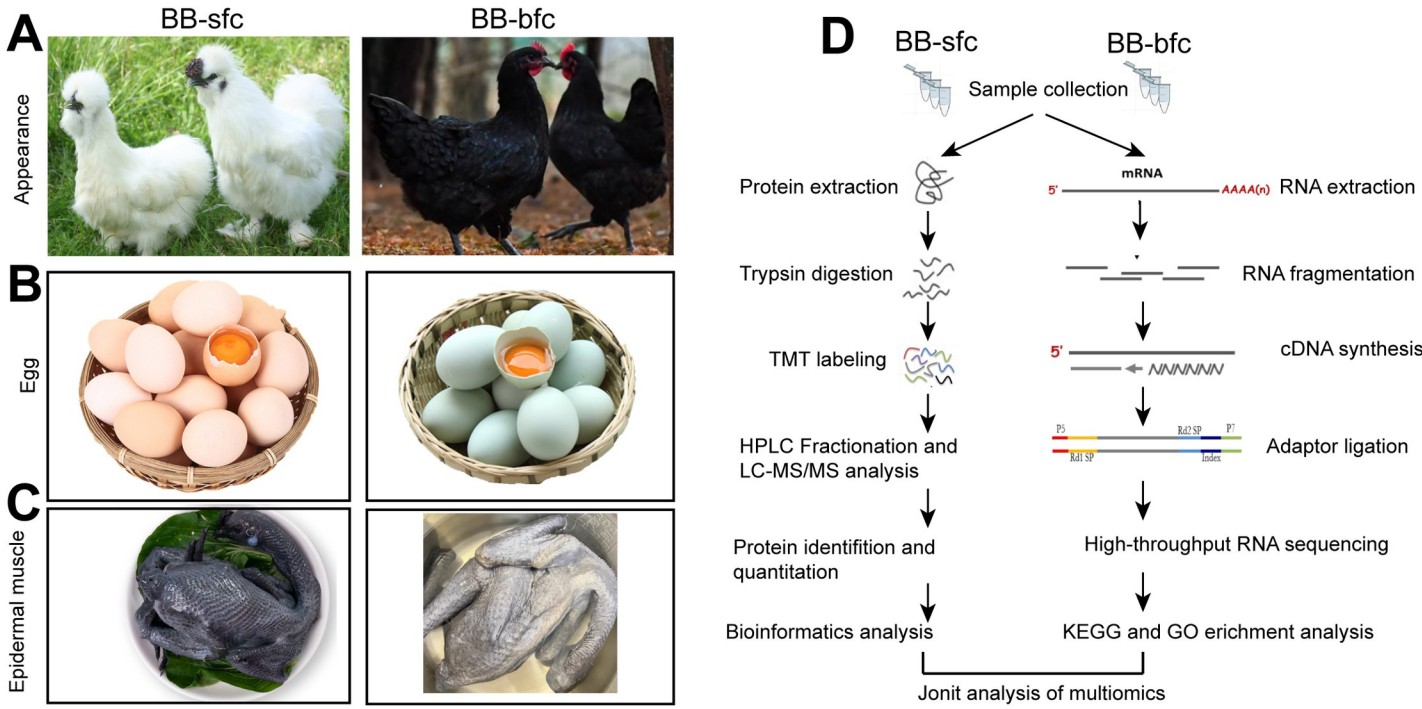

**Fig 1. Physiological differences and high-throughput sequencing strategy were presented in two Chinese Taihe Black-bone chickens.** (A) The appearance characteristics of Black-Bone silk fowl chicken (BB-sfc) and Black-Bone and black-feathered chicken (BB-bfc). (B) Representative images of egg differences between BB-sfc and BB-bfc chickens. (C) Representative images of muscle differences between BB-sfc and BB-bfc chickens. (D) Workflow of high-throughput transcriptome and quantitative proteome analysis of chicken muscle using RNA-Seq and TMT-based LC-MS/MS methods.

genes and identify the key muscle development genes in Taihe silky fowl chicken, we also constructed the PPI interaction network by using the STRING database. The results suggested that 36 genes were significantly enriched in network analysis (Fig 2D). For example, autophagy related 14 (ATG14) interacted tightly with syntaxin 17(STX17) and PRKAB1; genes related to protein synthesis and degradation including WD repeat domain 36 (WDR36), and nucleolar protein 6 (NOL6) were also showing a significant interaction relationships between each other. Together, our findings showed that Taihe BB-sfc chicken significantly differ in their levels of gene transcription compared to the local BB-bfc chicken.

### 3.3 GO and KEGG pathway enrichment analysis of Taihe silky fowl in RNA-Seq

In order to better understand the process of muscle development, we performed the GO enrichment analysis on the DEGs between BB-sfc and BB-bfc chicken breeds. The results demonstrated

**Table 1. Basic statistical characteristics of RNA sequencing in chicken muscle samples.**

| Sample | Raw reads | Clean reads | Mapped reads | Q20(%) | Q30(%) | GC (%) |
|---|---|---|---|---|---|---|
| BB-bfc_rep1 | 60,652,534 | 60,286,038 | 50,127,468(83.15%) | 96.77 | 91.27 | 53.62 |
| BB-bfc_rep2 | 66,831,426 | 66,431,206 | 54,588,627(82.17%) | 96.05 | 89.54 | 53.38 |
| BB-bfc_rep3 | 53,417,142 | 53,082,682 | 44,397,448(83.64%) | 97.22 | 92.51 | 53.61 |
| BB-sfc_rep1 | 74,377,526 | 73,930,666 | 62,266,455(84.22%) | 96.52 | 90.66 | 53.37 |
| BB-sfc_rep2 | 63,726,236 | 63,343,654 | 53,699,709(84.78%) | 96.4 | 90.37 | 53.26 |
| BB-sfc_rep3 | 58,607,642 | 58,251,520 | 49,968,972(85.78%) | 96.83 | 91.44 | 52.62 |

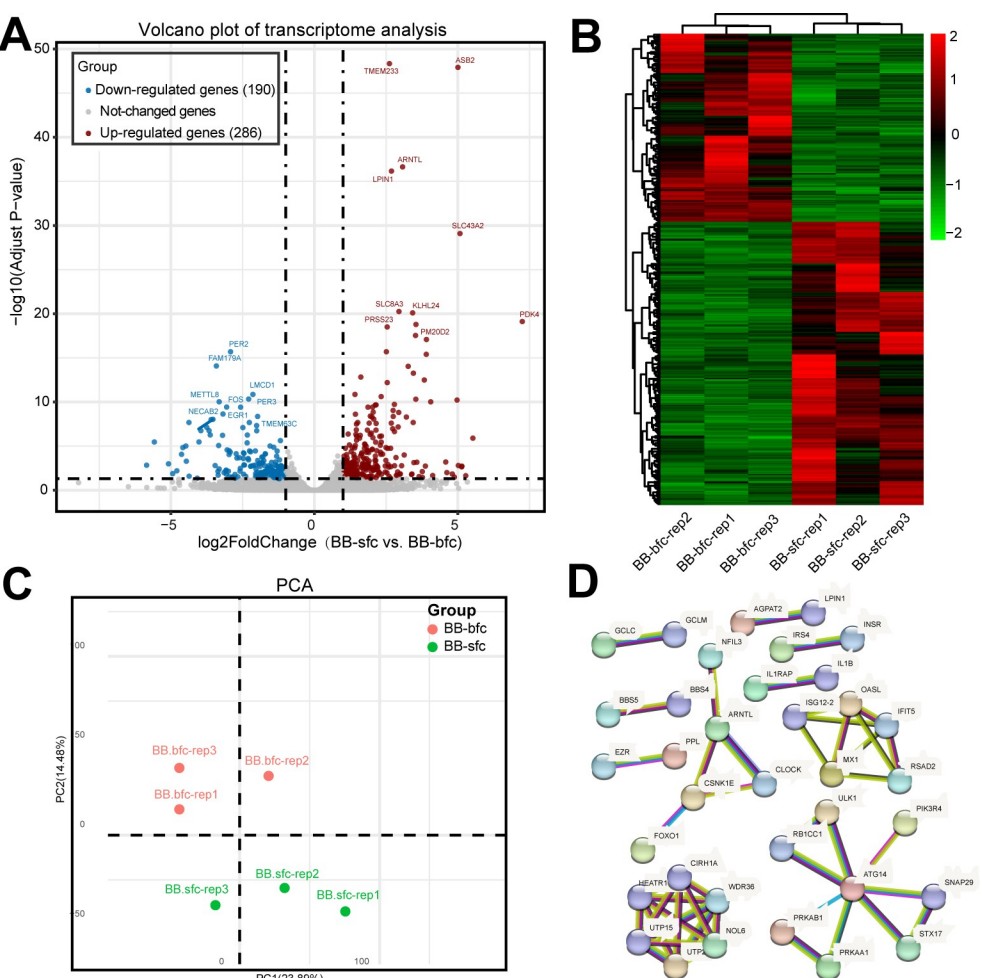

**Fig 2. The differentially expressed genes of the Taihe silky fowl chicken in muscle samples were identified by transcriptome analysis.** (A) The volcano plot represented the DEGs identified in the BB-sfc compared with BB-bfc chicken breeds using the DESeq2 method. The X-axis means the change of gene multiplicity (log$_2$ FoldChange), and the Y axis indicates the significance level of the difference (-log$_{10}$ Adjust P-value). The red and blue dots showed the significantly up- and down-regulated DEGs, respectively. (B) The heatmap of DEGs between BB-sfc and BB-bfc breed chickens was performed by hierarchical clustering analysis. (C) RNA libraries of BB-sfc and BB-bfc presented obvious differences by two-dimensional PCA analysis. (D) The integrated network presented the interaction relationship of DEGs in Taihe Black-bone silky fowl chicken.

that cellular respiration, generation of precursor metabolites and energy in BP terms; inner mitochondrial membrane protein and respiratory chain complex in CC terms; oxidoreductase activity and electron transfer activity in MF terms were significantly enriched in up-regulated differentially genes in Taihe black-bone silky fowl compared with BB-bfc chickens (Fig 3A). Besides, phospholipid homeostasis, response to stress in BP terms; cholesterol transporter activity in MF terms, Atg1/ULK1 kinase complex, SCF ubiquitin ligase complex in CC terms were significantly enriched in down-regulated genes in BB-sfc chickens (Fig 3B).

To gain more insight into the process of muscle development, we performed the KEGG pathway analysis on the DEGs between the BB-sfc and BB-bfc chickens. Our findings indicated that the top three key enriched signaling pathways in the up-regulated genes of Taihe black-bone silky chicken were oxidative phosphorylation, metabolic pathways, and citrate cycle (Fig 3C). However, PPAR signaling pathway, FOXO signaling pathway and animal autophagy were

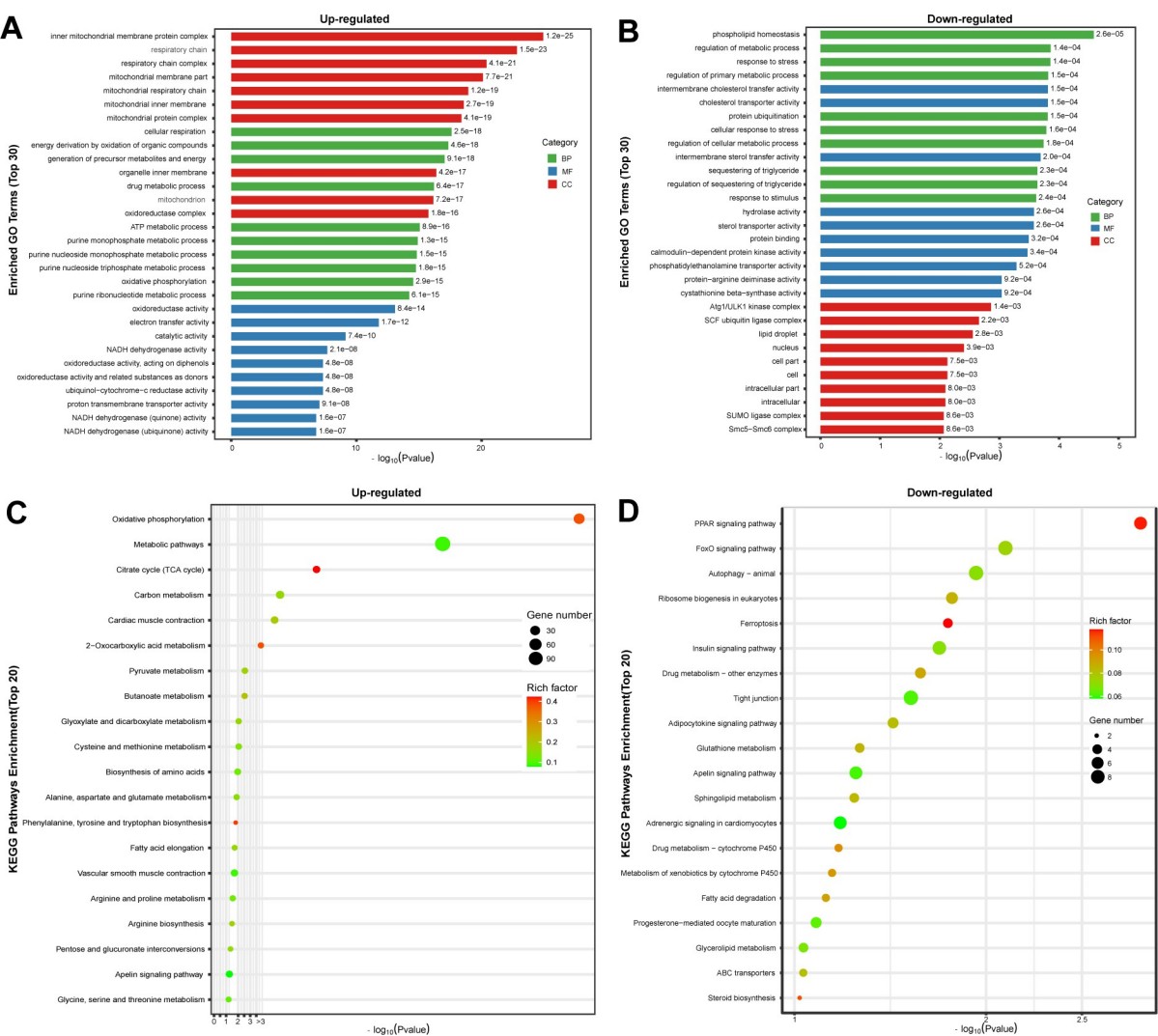

**Fig 3. GO enrichment and KEGG pathway analysis of up- and down-regulated genes in Taihe silky fowl chicken using RNA-Seq.** (A) The GO enrichment bar graphs for the top 30 items in the up-regulated genes in Taihe silky fowl chicken. (B) The GO enrichment bar graphs for the top 30 items in the down-regulated genes in Taihe silky fowl chicken. Abbreviations: biological processes (BP), cellular components (CC) and molecular functions (MF). (C) The 20 most significant KEGG pathways were identified in significantly up-regulated genes between BB-sfc and BB-bfc chickens. (D) The 20 most significant KEGG pathways were identified in significantly down-regulated genes between BB-sfc and BB-bfc chickens.

the top three enriched pathways in the down-regulated genes of Taihe black-bone silky fowl chicken (Fig 3D).

## 3.4 Diverse protein profiling between Taihe BB-sfc and BB-bfc chickens

To further investigate protein dynamic changes between different chicken breeds, we performed chicken muscle proteomic analysis between BB-sfc and BB-bfc chickens using TMT-based quantitative method. Totally, 12, 795 peptides and 10, 956 unique peptides were identified. Among which, 2,119 proteins were identified and 2,117 were quantifiable in all chicken samples (Fig 4A and S3 Table). In order to explore the protein difference of chicken muscle with breeds, we determined the differentially expressed proteins (DEPs) between two groups and p < 0.05 along with |fold change|>1.2 was used as the filtering criteria for DEPs. The results suggested

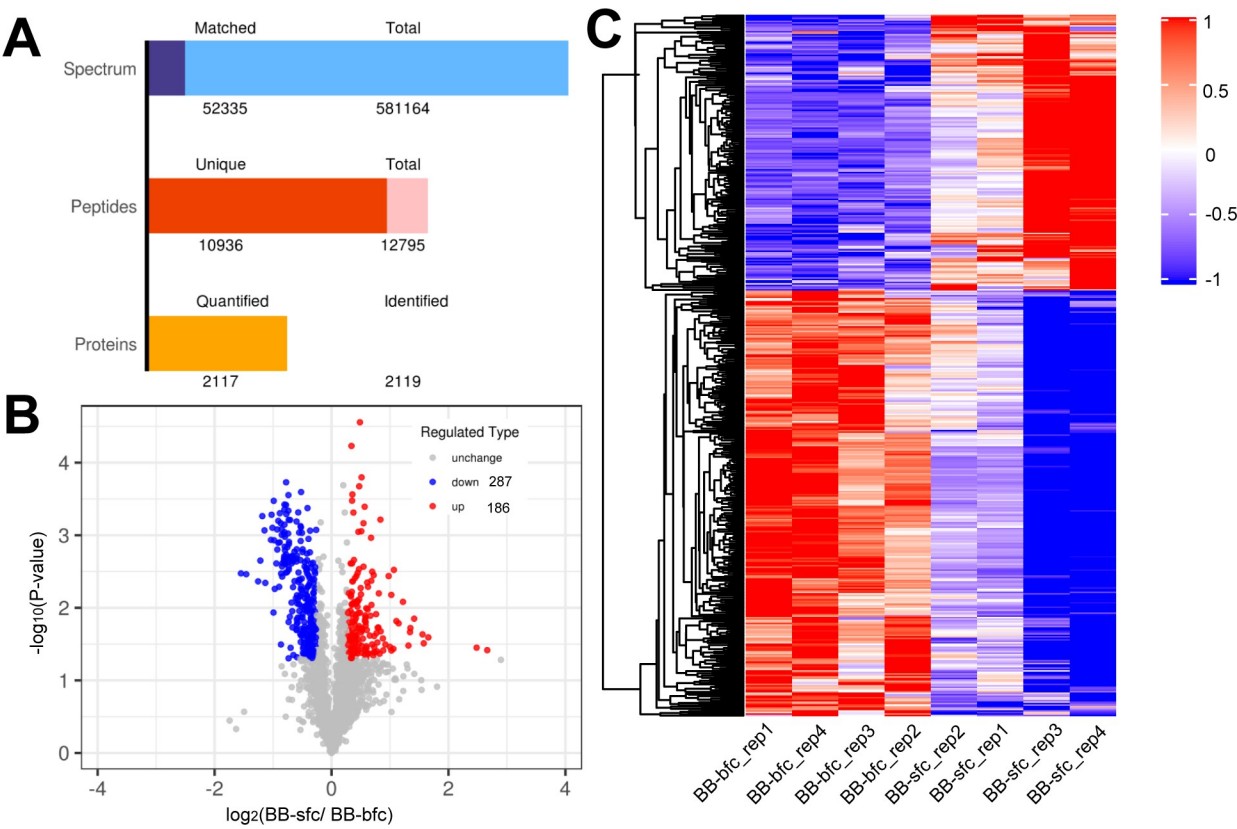

**Fig 4. The differential protein screening in Taihe silky fowl chicken using the TMT-based proteomic analysis.** (A) Protein identification and quantitative result statistics in proteomic analysis. (B) Volcano map showing the significant proteins differences on muscles between BB-sfc and BB-bfc chicken breeds. (C) Profiles of different expressed proteins in Taihe silky fowl chicken using the hierarchical clustering analysis. The red presents the significantly up-regulated and blue presents the down-regulated proteins, respectively.

that 186 were significantly up-regulated and 287 were down-regulated proteins in BB-sfc chickens compared with the BB-bfc chicken breeds (Fig 4B). The hierarchical clustering analysis of the DEPs in the two comparsions revealed that most of DEGs were down-regulated DEPs and the gene expression trends between four biological replicates were relatively similar (Fig 4C).

## 3.5 Functional enrichment analysis of DEPs in BB-sfc and BB-bfc chickens

Sub-cellular organelles such as mitochondria, endoplasmic reticulum have the certain shape and function in the cytoplasm, which are important places where proteins switch their functions. Different sub-cellular organelles often perform different cellular functions, so analyzing the sub-cellular location of proteins is conducive to further exploring the functions of proteins in cells. We used the subcellular structure prediction software CELLO to perform subcellular localization analysis on all differentially expressed proteins. The results suggested that most of the DEPs were classified into the cytoplasmic (172), mitochondrial (162) and nuclear (142) in the cells (Fig 5A). Next, the possible protein-protein interactions of differentially abundant proteins were analyzed based on the STRING database using the CytoScape software. The results demonstrated that NADH: ubiquinone oxidoreductase core subunit such as NDUFB5, NDUFB9, NDUFA10 and NDUFS8 exhibited the tightly protein-protein interactions and closed related in function. Furthermore, citrate synthase (cs), pyruvate dehydrogenase beta (PDHB) and fumarate hydratase (fh) also demonstrated the strongest interactions in chicken muscles (Fig 5B).

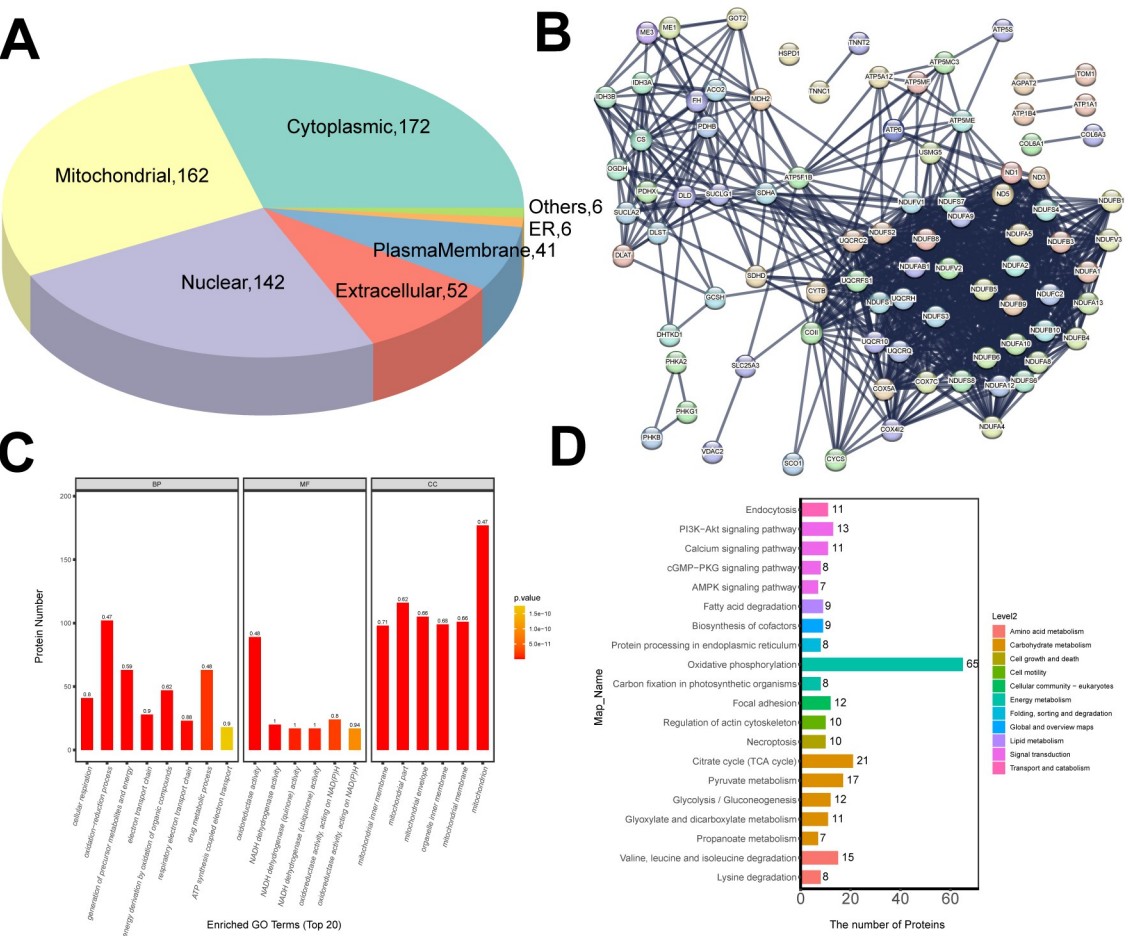

**Fig 5. The functional enrichment and network analysis of DEPs in Taihe silky fowl chicken.** (A) Pie chart shows the number and distribution proportion of proteins in each sub-cellular organ. (B) Interaction networks of differentially abundant proteins in Taihe silky fowl. (C) Representative GO annotation of differentially abundant proteins for biological processes, molecular functions and cellular components, respectively. (D) KEGG pathway enrichment analysis. The vertical axis represents KEGG pathways and the horizontal axis represents the number of differentially expressed proteins in each pathway.

We use Blast2GO software to perform GO function on all differentially expressed proteins in Taihe silky fowl chicken. Our results suggested that oxidation-reduction process and drug metabolic process were significantly enriched BP terms in Taihe silky fowl chicken, while oxidoreductase activity in MF terms and mitochondrion in CC terms were also significantly enriched in BB-sfc chickens (Fig 5C). Besides, in order to comprehensively analyze biological processes, disease occurrence, and drug action mechanisms, it is often necessary to elucidate the changes from the perspective of a series of protein coordination, such as metabolic pathway development. Therefore, we annotate differential protein analysis through the KEGG analysis and oxidative phosphorylation was the top enriched, followed by the citrate cycle and pyruvate metabolism (Fig 5D). Taken together, these results further demonstrated that Taihe silky fowl chicken had stronger antioxidant capacity in muscles compared to the ordinary black-feathered chickens.

## 3.6 Integrated transcriptome and proteome analysis in Taihe silky fowl chicken

To intersect the correlation of transcriptome and proteome data, the DEPs and DEGs were compared between the BB-sfc and BB-bfc chicken breeds. Of the up-regulated DEGs and

DEPs, a total of 6 overlapped genes (CBFB, POSTN, AGPAT2, CRP, METTL7A and MTPN) were identified among the two groups (Fig 6A). Glycerophospholipids are the major component of cell membranes and are involved in chemical signaling within cells. Triacylglycerols are fat molecules that are stored in adipocytes for later conversion to energy. On the other hand, of the down-regulated DEGs and DEPs, a total of 8 overlapped genes (CYTB, AMPD1, CARNS1, CKMT2, PCCB, DUSP26, IDH3A and FN3K) were identified among the two groups (Fig 6B). Our results showed that the mRNA and protein levels of CARNS1 were significantly decreased in Taihe silky fowl chickens (Fig 6C). Furthermore, genes related to mitochondria and antioxidant activity such as CKMT2 (creatine kinase, mitochondrial 2) and IDH3A (isocitrate dehydrogenase 3 alpha) were also greatly down-regulated of transcripts and proteins in Taihe silky fowl chicken. All-together, these results further demonstrated that Taihe black-bone silky fowl chicken showing unique characteristics of muscles tissues in mitochondrial energy metabolism and antioxidant capacity.

## 4. Discussion

The Chinese Taihe silky fowl chicken is a national product of geographical indication that known for its top ten characteristics, high nutritional and medicinal value [28, 29]. The chicken of Taihe Black-bone silky fowl is fresh and tender in meat with high protein and low fat, which the average crude protein content of 52.72% and crude fat only accounting for 24.17% [30]. The nutritional components of meat affects chicken quality and economic value, thus it is very necessary to reveal the molecular mechanism of breast muscle development in BB-sfc chicken. Previous studies have uncovered several genes and signaling pathways involved in chicken muscle fibers development and breast muscle growth in different breeds of chickens [31, 32]. Our previous studies have identified the metabolic alterations between Taihe silky fowl chicken and black-feathered chicken by performing an un-targeted UHPLC-Q-TOF-MS/MS analysis [33]. However, there have been few studies investigating the proteomic changes and molecular mechanisms involved in muscle fiber development in Taihe

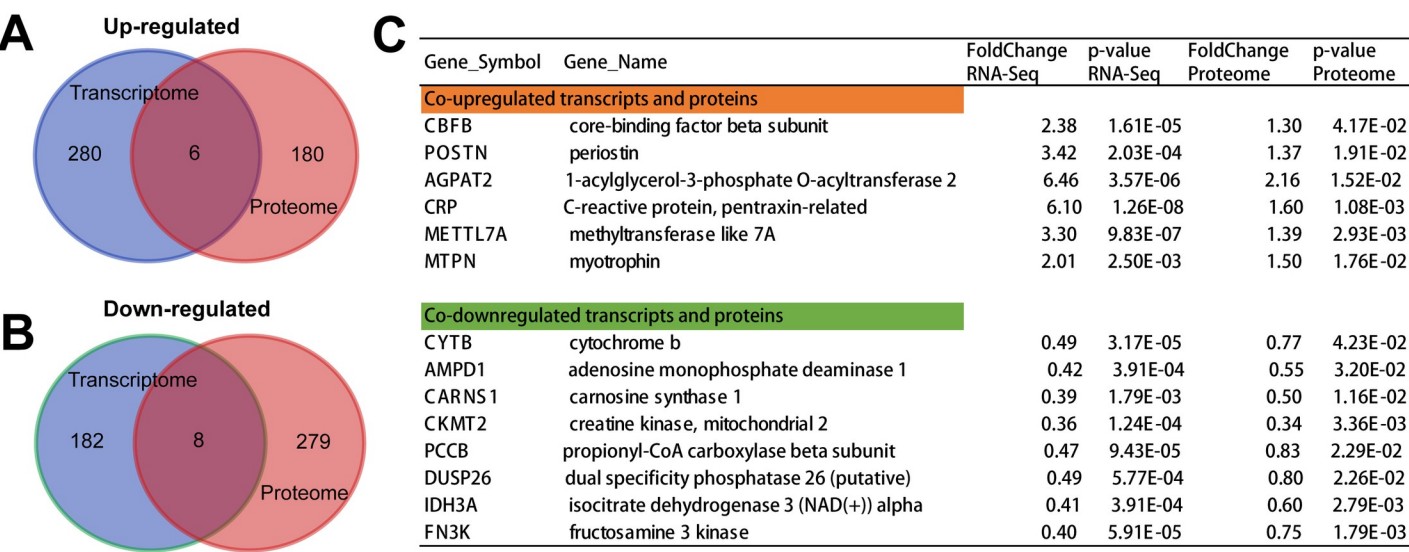

**Fig 6. The integrated transcriptome and proteome analysis identified the specific muscle components in Tailhe silky fowl chicken.** (A) Venn diagram of up-regulated differentially expressed transcripts and proteins in Taihe silky fowl chicken. (B) Venn diagram of down-regulated differentially expressed transcripts and proteins in Taihe silky fowl chicken. (C) The statistical table in the fold-change and significance differences in genes and proteins that commonly up-regulated and down-regulated on the muscle tissues in Taihe silky fowl chicken.

silky fowl chicken. In order to gain a better understanding of the mechanism of breast muscle development, we utilized advanced technologies such as TMT-based quantitative proteomics and RNA-Seq to analyze the protein and mRNA profiles in breast muscles of BB-sfc and BB-bfc chickens.

In the current study, RNA-Seq analysis revealed that 286 up-regulated and 190 down-regulated genes in BB-sfc chicken compared to the BB-bfc chicken. Previous studies suggested that the process of autophagy mediated by Atg family is crucial in delaying skeletal muscle aging [34]. Atg genes play a regulatory role in autophagy, either facilitating or inhibiting the process to enhance the physiological structure and function of skeletal muscle, and specifically, ATG14 has vital functions in maintaining muscle homeostasis [35]. The integrated network analysis suggested that 36 genes such as ATG14 were substantially enriched in Taihe silky fowl chickens, which were denoted as candidate genes for breast muscle growth. Based on the KEGG pathway and GO enrichment analysis, the Taihe silky fowl chicken showed significant enrichment in the oxidative phosphorylation and citrate cycle. Our results are consistent with previous findings that mitochondrial content is considered a valuable characteristic for distinguishing muscle fiber types in terms of energy metabolism in breast muscles [36]. Previous research has examined and analyzed the three stages of chicken muscle tissue development using RNA sequencing. The analysis revealed a detection of 978 genes with differential expression and these genes primarily participated in cellular growth, development of muscle tissue, and cellular differentiation and proliferation [37]. Intramuscular fat content is a crucial aspect of meat quality, and PLIN2 can be regarded as a molecular marker for assessing the quality of poultry meat [38]. Transcriptomics analysis of Daheng Bruners chicken reveals that PLIN2 plays a role in regulating chicken muscle growth, differentiation, and apoptosis [39]. However, our research indicated that there is no significant difference in the expression of the PLIN2 gene between BB-sfc and BB-bfc chickens, suggesting that this gene exhibits different expression characteristics in chicken muscle development.

Due to the complexity of post-transcriptional control mechanisms in organisms, gene expression is not always consistent at the transcriptional and translational levels [40]. To complementarity of data, we performed TMT-based quantitative proteomics analysis on the same batch of samples. According to our findings, 287 proteins were considerably down-regulated while 186 proteins were significantly up-regulated in BB-sfc chickens compared to BB-bfc chicken breeds. It is suggested that Taihe silky fowl chicken had strongly enriched in drug metabolism and the oxidation-reduction process. From our results, it is further demonstrated that the mitochondrial function and oxidative phosphorylation in Taihe silky fowl muscle were significantly increased, while cholesterol content and lipid metabolism were significantly decreased. Our results support previous research that insulin acutely enhances the mitochondrial function of both rat and human skeletal muscle by increasing the coupling efficiency of oxidative phosphorylation [41]. Through real-time measurements, it has been observed that insulin rapidly improves the coupling efficiency of oxidative phosphorylation and enhances cellular respiratory control [42]. Besides, embryonic growth and development of skeletal muscle play a crucial role in determining muscle mass and significantly impact meat production in chickens [43]. In the previous studies, the proteomics quantification analysis using iTRAQ method was conducted on muscle tissues of female chickens. And protein interaction network analyses revealed that differentially expressed proteins primarily participate in pathways related to protein synthesis, muscle contraction, and oxidative phosphorylation [44, 45]. The differential proteins identified in Taihe black-bone chickens, although having a low overlap rate with transcriptome data, were consistent with these previous research data.

To comprehensive analysis of transcriptome and proteome data, the DEPs and DEGs were compared between the BB-sfc and BB-bfc chickens. The results suggested that a total of 6

overlapped genes (CBFB, POSTN, AGPAT2, CRP, METTL7A and MTPN) were identified among the two chickens. Besides, among the down-regulated DEGs and DEPs, a total of 8 overlapped genes (CYTB, AMPD1, CARNS1, CKMT2, PCCB, DUSP26, IDH3A and FN3K) were identified in the Taihe silky fowl chickens. The AGPAT2 gene can cause changes in the function of glycerol phosphoesterase, thereby affecting the normal development and differentiation of adipocytes [46]. CARNS1, also known as carnosine synthase 1, belongs to the ATP-grasp family of ATPases and its main function is to facilitate the generation of carnosine and predominantly present in skeletal muscle tissue [47]. The low consistency ratio between the transcriptome and proteome of Taihe silky fowl chicken may be due to significant differences in the regulatory processes of transcription and translation. Moreover, our results showed that the mRNA and protein levels of mitochondria and antioxidant activity such as CKMT2 and IDH3A were also greatly down-regulated both transcripts and proteins in Taihe silky fowl chicken. Therefore, we need to verify the functional mechanisms of these identified candidate genes in muscle development of Taihe black-bone chickens through more molecular experiments and pharmacological studies in the near future. Taken together, our findings collectively demonstrated that Taihe black-bone silky fowl chicken exhibited unique traits in mitochondrial energy metabolism and antioxidant capability on breast muscle tissues.

## 5. Conclusion

In summary, our study used RNA-Seq and proteomics to analyze gene expression and protein changes between two breeds of Taihe black-bone chickens. RNA-Seq and proteome analysis revealed many differentially genes or proteins, mostly affecting mitochondrial function and oxidative phosphorylation, which are crucial for muscle antioxidant capacity. Meanwhile, multi-omics analysis revealed that dozens of genes are co-expressed in the transcriptome and proteome, which were found to be linked to improved muscle antioxidant status. Altogether, this research provides a valuable database for understanding the unique traits of Taihe Black-Bone silky fowl chickens, and molecular functions of these genes in chicken muscle tissue development is awaiting for further investigation.

## Supporting information

**S1 Fig. The average egg weights of BB-sfc and BB-bfc chickens were shown.**
(TIF)

**S2 Fig. Graphical abstract.**
(TIF)

**S1 Table. The growth parameters of two breeds chickens were calculated in this study.**
(XLSX)

**S2 Table. The differential genes were identified in the silky fowl chicken by RNA-Seq.**
(XLSX)

**S3 Table. The differentially proteins were identified by TMT-based proteome analysis.**
(XLSX)

**S4 Table. Quality detection statistics of RNA sequencing libraries from two breeds of black-bone chickens.**
(XLSX)

## Acknowledgments

We would like to give special thanks to the Jiangxi Wangbeitu Taihe silky fowl Development Co., Ltd. and Jiangxi Aoxin Taihe silky fowl Development Co., Ltd. for providing the sequencing samples.

## Author Contributions

**Conceptualization:** Guanghua Xiong, Xinjun Liao.

**Data curation:** Wanqing Chen, Kai Jiang, Shuyuan Liu, Juan Li.

**Formal analysis:** Wanqing Chen, Shuyuan Liu, Juan Li.

**Funding acquisition:** Xinjun Liao.

**Investigation:** Kai Jiang.

**Methodology:** Wanqing Chen, Shuyuan Liu.

**Project administration:** Guanghua Xiong.

**Supervision:** Xinjun Liao.

**Validation:** Kai Jiang, Juan Li.

**Writing – original draft:** Guanghua Xiong.

**Writing – review & editing:** Guanghua Xiong.

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
