## [Decision Letter · Decision Letter 0]

29 Nov 2023

PONE-D-23-37042Integrated transcriptome and proteome analysis reveals the unique nutritional composition on the muscles in Chinese Taihe Black-bone Silky Fowl (Gallus gallus domesticus Brisson)PLOS ONE

Dear Dr. Liao,

Thank you for submitting your manuscript to PLOS ONE. After careful consideration, we feel that it has merit but does not fully meet PLOS ONE’s publication criteria as it currently stands. Therefore, we invite you to submit a revised version of the manuscript that addresses the points raised during the review process.

We look forward to receiving your revised manuscript.

Kind regards,

Tomoyoshi Komiyama, Ph.D

Academic Editor

PLOS ONE

Journal Requirements:

In addition, we noticed you have some minor occurrence of overlapping text with the following previous publication(s), which needs to be addressed:

- https://medlineplus.gov/genetics/gene/agpat2/

- https://www.ncbi.nlm.nih.gov/gene/136319

In your revision ensure you cite all your sources (including your own works), and quote or rephrase any duplicated text outside the methods section. Further consideration is dependent on these concerns being addressed.

"This work was financially supported by the National Natural Scientific Foundation of China (82160048), Jiangxi Natural Science Foundation Project (20202ACBL215009) and Jiangxi Science and Technology Research Project of the Education Department(GJJ2201611)."

Additional Editor Comments:

Dear Authors,

Your research analyzed the unique nutritional composition of the muscles in Chinese Taihe Black-bone Silky Fowl.

For the research, you performed a comprehensive analysis of the muscle transcriptome and proteome in order to investigate the molecular mechanisms underlying chicken growth.

You suggested that these findings collectively support the concept that the Taihe black-bone cocoon chicken exhibits unique properties in mitochondrial energy metabolism and muscle tissue antioxidant capacity.

In the attached reviews, the reviewers have given feedback your manuscript.

I have carefully considered your manuscript.

Unfortunately, I decided major revision is required based on the three reviewer’s comments.

I think that it is necessary to strengthen the reliability of your results by adding as much information as possible.

I believe these comments will be very helpful in the revision of your study.

If your results become clear, I think your research will be important for the future of any researchers who aim to better understand this subject.

In its current state, I am unable to accept this manuscript. After revision, please feel welcome to resubmit.

Thank you for your understanding.

I have an additional comment.

Please include the results of the RNA sample quality test in the methods or supplemental section to prove the reliability of the data.

For example, nanodrop and bioanalyzer tests.

Tomoyoshi Komiyama

Reviewers' comments:

Reviewer's Responses to Questions

**Comments to the Author**

1. Is the manuscript technically sound, and do the data support the conclusions?

Reviewer #1: No

Reviewer #2: Partly

Reviewer #3: Yes

2. Has the statistical analysis been performed appropriately and rigorously? 

Reviewer #1: Yes

Reviewer #2: Yes

Reviewer #3: Yes

3. Have the authors made all data underlying the findings in their manuscript fully available?

Reviewer #1: Yes

Reviewer #2: No

Reviewer #3: Yes

4. Is the manuscript presented in an intelligible fashion and written in standard English?

Reviewer #1: No

Reviewer #2: No

Reviewer #3: Yes

5. Review Comments to the Author

Reviewer #1: Review of PONE-D-23-37042, Integrated transcriptome and proteome analysis reveals the unique nutritional composition on the muscles in Chinese Taihe Black-bone Silky Fowl (Gallus gallus domesticus Brisson) by Guanghua Xiong et al.

The manuscript describes results of the integrated analysis of the transcriptome and proteome of muscles of two native Chinese chicken breeds with unique nutritional values. In itself such studies are interesting, even when the breeds themselves are of limited (commercial) value.

Major comments

1. The present manuscript lacks language clarity and should be revised. Usually, I provide a list of language suggestions, but here the list would have been 10 pages or so…… So, I suggest to contact a language editing company to improve the English presentation of the results.

2. The Discussion section needs extensive rewriting, basically to the level of setting up an entire new Discussion section. Part of the results section need to be moved to the Discussion section, the results need to be removed from the Discussion section, and conclusions need to be drawn from the results after discussing them with the existing literature. Most of these points are either lacking from the present Discussion section, or the readability is such that I did not get it.

Minor comments

Line (L)17: nutritional components: What do you mean? What the chicken eats? What the nutritional value is for consumers? Please be specific.

L18: Biomarkers for what?

L19: Growth: general growth, or muscle development in specific?

L23: laid green-shelled eggs: only later it became known to me that this is possibly a breed, it is unclear here. Just add “chicken” or “breed” and the reader is not confused anymore.

L35: Overlapped: what do you mean by overlapped? transcriptome and proteome? What about the others? Are they false positives?

L62-64: Please rewrite sentence

Section 3.1 seems to contain no experimental results of this study. correct? Please delete.

L231-235: This is Materials ands Methods. Please delete here. If necessary, move to the Materials and Methods section.

L264-267: Not in the Results section

L283-285: Move to the Discussion section.

L301: micro-organs? Do you mean organelles?

L336-339: Move to the Discussion section.

L346-348: Move to the Discussion section.

L372-278: What is the meaning of this? It is so unclear to me that I cannot make a suggestion what to do with it.

L398-400: Repetition of results is no discussion. What are the meanings of your results? Please discuss that.

References: no given names of authors

Reviewer #2: General comments: The authors have put together an interesting data on the muscle of Chinese Taihe Black-bone Silky Fowl. However, there are some modifications to be made to this manuscript before is suitable for publication.

1. Line1-2: What is the unique nutritional composition of muscle in Chinese Taihe Black-bone Silky Fowl? I can not find that in the manuscript.

2. Line 23: What does ‘Black Feathered chicken and laid green-shelled eggs’ mean? Black Feathered chicken? laid green-shelled eggs?

3. Line 23: Why was Black Feathered chicken selected for transcriptome and proteome analysis? Which breed does Black Feathered chicken belong to?

4. Line 102: abdominal muscle tissues?

5. Line 115-116: ‘all the chickens were randomly selected’ should be modified to ‘all the chickens were selected’

6. Line 117: How were the chicken euthanized?

7. Line 201-202: Which statistical analyses in the study were performed using the SPSS 20.0 software?

8. Line 214-215: where was the data on eggs obtained from?

9. Line 218: What does ‘data not shown’ mean? Is that your result?

10. Line 221-224: Why did proteomic quantitative analysis was performed using four biological replicates of muscle samples but RNA-Seq three biological replicates?

11. Which part of result does the Figure 7 belong to? I did not find Figure 7 in the results and and legend of Figure 7 of manuscript.

12. Please upload your original sequencing data.

Reviewer #3: The manuscript is written clearly with the exception of some few areas that need to be addressed as follows:

1. there are several grammatical errors- L40-41, 62, 68, and112.

2. avoid repeating methods in results-L210-212, 231-233, and 367-374.

6. PLOS authors have the option to publish the peer review history of their article (what does this mean?). If published, this will include your full peer review and any attached files.

Reviewer #1: No

Reviewer #2: No

Reviewer #3: No

---

## [Author Response · Author response to Decision Letter 0]

26 Dec 2023

Reviewer #1: Major comments

1. The present manuscript lacks language clarity and should be revised. Usually, I provide a list of language suggestions, but here the list would have been 10 pages or so…… So, I suggest to contact a language editing company to improve the English presentation of the results.

Answer：Thank you for your nice suggestions. We have extensively revised and polished the manuscript through a professional language editing company, and all modified parts are marked in red in the manuscript

2. The Discussion section needs extensive rewriting, basically to the level of setting up an entire new Discussion section. Part of the results section need to be moved to the Discussion section, the results need to be removed from the Discussion section, and conclusions need to be drawn from the results after discussing them with the existing literature. Most of these points are either lacking from the present Discussion section, or the readability is such that I did not get it.

 Answer：Thank you for your good suggestions. We have rewritten the discussion section according to your suggestion and moved some of the results to the discussion section, while deleting the experimental results restatement in the discussion section.

Minor comments

Line (L)17: nutritional components: What do you mean? What the chicken eats? What the nutritional value is for consumers? Please be specific.

Answer：Thank you for your nice suggestions. Firstly, the nutritional components means the silky fowl chicken have higher nutritional substances in their muscles such as amino acids, trace elements, and unsaturated fatty acids. Secondly, the normal chicken diet mainly contains 60% corn, 20% soybean meal and 10% wheat bran. Thirdly, the nutritional value of anti-aging, anti-cancer, and immune enhancing functions for consumers. We have revised the manuscript in the abstract and method sections. 

L18: Biomarkers for what?

Answer：Thanks. Biomarkers means the differentially expressed genes and proteins in Taihe silky fowl chickens, which may serve as distinguishing markers between Taihe silky fowl chicken and other breeds of chickens.

L19: Growth: general growth, or muscle development in specific?

Answer：Thanks. In this article, we specifically refer to muscle development.

L23: laid green-shelled eggs: only later it became known to me that this is possibly a breed, it is unclear here. Just add “chicken” or “breed” and the reader is not confused anymore.

Answer：Thanks for you good suggestions. We will uniformly name “Taihe silky fowl” as “Black-Bone silky fowl chicken (BB-sfc)”, and “Black Feathered chicken and laid green-shelled eggs” as “Black-Bone black-feathered chicken (BB-bfc)”.

L35: Overlapped: what do you mean by overlapped? transcriptome and proteome? What about the others? Are they false positives?

Answer：Thanks for you nice suggestion. “Overlapped“ means that these genes and proteins appeared simultaneously in the transcriptome and proteome analysis. Perhaps there are false positives in this, and we will conduct experimental confirmation on these genes and proteins in future research.

L62-64: Please rewrite sentence

Answer：Thanks for your reminder. We have rewritten this sentence.

Section 3.1 seems to contain no experimental results of this study. correct? 

Answer：Thanks for your suggestions. Fig. 1A mainly aims to show the physiological differences between Taihe silk fowl chicken and local Black-feathered chicken, including differences in appearance and egg weight. Some data was provided in the supplementary materials. Fig. 1B mainly presents the main process transcriptome sequencing and proteome analysis.

L231-235: This is Materials ands Methods. Please delete here. If necessary, move to the Materials and Methods section.

Answer：Thanks for your suggestions. We have moved these sentences to the Materials and Methods section.

L264-267: Not in the Results section

Answer：Thanks for your suggestions. We have deleted these sentences.

L283-285: Move to the Discussion section.

Answer：Thanks for your reminder. We have moved it to the Discussion section.

L301: micro-organs? Do you mean organelles?

Answer：Thanks for your reminder. This word of “micro-organs” is indeed a bit ambiguous. We specifically refer to organelles. We have deleted this word.

L336-339: Move to the Discussion section.

Answer：Thanks for your reminder. We have moved it to the Discussion section.

L346-348: Move to the Discussion section.

Answer：Thanks for your reminder. We have moved it to the Discussion section.

L372-378: What is the meaning of this? It is so unclear to me that I cannot make a suggestion what to do with it.

Answer：Thanks for your suggestions. This sentence is indeed a bit ambiguous, and we have deleted it.

L398-400: Repetition of results is no discussion. What are the meanings of your results? Please discuss that.

Answer：Thanks for your suggestions. We have revised this section and discussed the meanings of our results.

References: no given names of authors

Answer：Thanks for your reminder. We have adopted the reference format specified by the PLOS ONE magazine.

Reviewer #2: General comments: The authors have put together an interesting data on the muscle of Chinese Taihe Black-bone Silky Fowl. However, there are some modifications to be made to this manuscript before is suitable for publication.

1. Line1-2: What is the unique nutritional composition of muscle in Chinese Taihe Black-bone Silky Fowl? I can not find that in the manuscript.

Answer：Thank you for your good suggestion. The Chinese Taihe Black-bone silky fowl chicken (BB-sfc) has high nutritional and medicinal value, rich in effective ingredients, and contains 18 amino acids higher than other chicken breeds[1]. According to previous literature reports, BB-sfc chicken is an alkaline food with high protein, low fat, and low sugar content, with an average crude protein content of 52.72% and crude fat only accounting for 24.17%, which the low fat ratio is comparable to soybeans and algae[2]. According to our RNA-Seq and proteome analysis, we identified 6 proteins (CBFB, POSTN, AGPAT2, CRP, METTL7A and MTPN) were up-regulated and 8 proteins (CYTB, AMPD1, CARNS1, CKMT2, PCCB, DUSP26, IDH3A and FN3K) were down-regulated in BB-sfc chickens. These identified proteins may be unique nutritional composition of muscle in Chinese Taihe Black-bone Silky Fowl.

[1] Liao X, Shi X, Hu H, Han X, Jiang K, Liu Y, et al. Comparative Metabolomics Analysis Reveals the Unique Nutritional Characteristics of Breed and Feed on Muscles in Chinese Taihe Black-Bone Silky Fowl. Metabolites. 2022;12(10).

[2] Zhou X, Liu L, Wang L, Liu T, Wu X. Proteomic study of Chinese black-bone silky fowl and the ring-necked pheasant egg white by iTRAQ technique. LWT. 2021;150:111936.

2. Line 23: What does ‘Black Feathered chicken and laid green-shelled eggs’ mean? Black Feathered chicken? laid green-shelled eggs?

Answer：Thank you for your good suggestion. We did not express this sentence clearly. Actually, there are many different breeds of Taihe black-bone chickens in China, among which the most typical is the black-bone silky fowl chicken (Namely BB-sfc). This breed of chicken has white feathers, black bones and meat, and lays eggs with red shells. Another breed is also a well-known local black-bone chicken, which we called the black-bone and black-feathered chicken (Namely BB-bfc). It has black feathers, black bones and meat, but the eggs it lays is green shells. The comparison pictures of the differences between these two breeds of chickens were shown as follows. We have made modifications to the corresponding sentences in the manuscript

3. Line 23: Why was Black Feathered chicken selected for transcriptome and proteome analysis? Which breed does Black Feathered chicken belong to?

Answer: Thank you for your question. The main objective of this article is to identify genetic diversity and differentially expressed genes and protein components in Taihe Black-Bone silky fowl chicken (BB-sfc) through proteomic and transcriptome analysis. The Black-Bone and black-feathered chicken (BB-bfc) is also a representative breed of Taihe black-bone chicken, so we chose it as a control with Taihe black-bone silky fowl chicken (BB-sfc).

4. Line 102: abdominal muscle tissues?

Answer：Thank you for your reminder. This statement is indeed a bit ambiguous. We are referring to “chicken breast meat”. We have made modifications to the relevant sentences in the revised manuscript. 

5. Line 115-116: ‘all the chickens were randomly selected’ should be modified to ‘all the chickens were selected’

Answer：Thank you for your suggestion. I have made the modifications to this sentence in the revised manuscript.

6. Line 117: How were the chicken euthanized?

Answer：Thank you for your question. The chickens were euthanized by using a firm percussive blow to the head followed by manual or mechanical cervical dislocation. We have added these statements in the 2.1 Materials and Methods section.

 7. Line 201-202: Which statistical analyses in the study were performed using the SPSS 20.0 software?

Answer：Thank you for your question. The statistical analysis using the SPSS 20.0 software is egg weight comparison, including body weight, feed intake, egg production, and feed conversion ratio between two chicken breeds.

8. Line 214-215: where was the data on eggs obtained from?

Answer：The data on eggs obtained from the average of 40 eggs between the two breeds of chickens, which is approximately 36.2 ± 3.1 g (BB-sfc) vs. 39.5 ± 4.3 g (BB-bfc). This data was also shown in Supplementary Fig. S1.

9. Line 218: What does ‘data not shown’ mean? Is that your result?

Answer：During raising these two breeds of chickens, we recorded many growth indicators including body weight, feed intake, egg production and feed conversion ratio etc., and we have included these data in Supplementary Table 3. Meanwhile, we deleted the words of “data not shown” in the revised manuscript.

10. Line 221-224: Why did proteomic quantitative analysis was performed using four biological replicates of muscle samples but RNA-Seq three biological replicates?

Answer：Thank you for nice suggestion. Based on previous research experience, transcriptome sequencing only requires three biological replicates per sample, but individual differences in samples of TMT-based proteome sequencing are larger than RNA-Seq. Therefore, in order to ensure the reliability of the results, we increased the number of biological replicates to four per sample in proteomic quantitative analysis.

11. Which part of result does the Figure 7 belong to? I did not find Figure 7 in the results and and legend of Figure 7 of manuscript.

Answer：Thank you for your reminder. Figure 7 is the Grapgh Abstract (Striking image) of the entire article, which should have appeared separately on the online version of the paper's webpage, so it was not included in the manuscript of the main text.

12. Please upload your original sequencing data.

Answer：Thank you for your reminder. The transcriptomics datasets generated during the current study are available in the NCBI SRA (PRJNA1051302). The mass spectrometry proteomics data have been deposited to the ProteomeXchange Consortium (http://proteomecentral.proteomexchange.org) via the iProX (Integrated proteome resources, http://www.iprox.org/) partner repository with the dataset identifier PXD047777. We have revised this information in the section of “data availability statement”.

Reviewer #3: The manuscript is written clearly with the exception of some few areas that need to be addressed as follows:

1. there are several grammatical errors- L40-41, 62, 68, and 112.

Answer：Thank you for your suggestions. We have made the following modifications to these sentences.

L40-L41. “Taken together, this research provides new insights into the molecular mechanism of antioxidant capacity in chicken muscles, and our study will laid the scientific basis for the improvement of the growth performance in Taihe black-bone silky fowl.” 

Reviesed to “Taken together, our studies provides a large database of gene expression and protein information of Taihe Black-Bone silky fowl chicken in muscles, which will provide a theoretical basis for fully exploring the unique economic value of Taihe BB-sfc chickens in the future.”

L62: “Due to traditional breeding require long-term selection and usually time-consuming, molecular breeding based on high-throughput sequencing can greatly improve breeding efficiency and shorten breeding cycles”

Revised to “Since conventional breeding requires long-term selection and generally takes a long time, the modern molecular genetic breeding using on high-throughput sequencing can significantly improve breeding efficiency and shorten the breeding cycle.”

L68: “which can be used to study gene expression and gene function prediction”

Revised to “which can be used to study the differentially expressed genes and regulated signaling pathways”.

L112: “the chickens of Black-feathered chicken and laid green-shelled eggs (BF-gsc)”

Revised to “Black-Bone and black-feathered chicken (BB-bfc)”

2. avoid repeating methods in results-L210-212, 231-233, and 367-374.

Answer：Thank you for your suggestions. We have deleted the corresponding sentences in L210-212, 231-233, and 367-374.

---

## [Decision Letter · Decision Letter 1]

22 Jan 2024

PONE-D-23-37042R1Integrated transcriptome and proteome analysis reveals the unique molecular features and nutritional components on the muscles in Chinese Taihe black-bone silky fowl chickenPLOS ONE

Dear Dr. Liao,

Thank you for submitting your manuscript to PLOS ONE. After careful consideration, we feel that it has merit but does not fully meet PLOS ONE’s publication criteria as it currently stands. Therefore, we invite you to submit a revised version of the manuscript that addresses the points raised during the review process.

We look forward to receiving your revised manuscript.

Kind regards,

Tomoyoshi Komiyama, Ph.D

Academic Editor

PLOS ONE

Additional Editor Comments:

Dear authors,

Thank you for submitting your revised manuscript.

I think it is easier to understand than the previous manuscript.

However, it received some additional questions from a reviewer.

Please answer these questions.

Tomoyoshi Komiyama

Reviewers' comments:

Reviewer's Responses to Questions

**Comments to the Author**

1. If the authors have adequately addressed your comments raised in a previous round of review and you feel that this manuscript is now acceptable for publication, you may indicate that here to bypass the “Comments to the Author” section, enter your conflict of interest statement in the “Confidential to Editor” section, and submit your "Accept" recommendation.

Reviewer #2: All comments have been addressed

Reviewer #3: All comments have been addressed

Reviewer #4: (No Response)

2. Is the manuscript technically sound, and do the data support the conclusions?

Reviewer #2: Yes

Reviewer #3: Yes

Reviewer #4: No

3. Has the statistical analysis been performed appropriately and rigorously? 

Reviewer #2: Yes

Reviewer #3: Yes

Reviewer #4: Yes

4. Have the authors made all data underlying the findings in their manuscript fully available?

Reviewer #2: Yes

Reviewer #3: Yes

Reviewer #4: Yes

5. Is the manuscript presented in an intelligible fashion and written in standard English?

Reviewer #2: Yes

Reviewer #3: Yes

Reviewer #4: Yes

6. Review Comments to the Author

Reviewer #2: The authors have put together an interesting data on the muscle of Chinese Taihe Black-bone Silky Fowl.

Minor comment:

1.Line140-141: |fold changes| or |log2(fold change)| ?

Reviewer #3: The manuscript has been revised adequately. All the issues raised have been adequately addresses including depositing of data in the NCBI.

Reviewer #4: In your research, you performed RNA-Seq, transcriptome, and proteome analyses using Black-Bone silky fowl chicken.　You identified genes from these analyses that were associated with improving the antioxidant status of muscles.　However, I have some suggestions.

1. Please make the abstract section a little more concise.

2. Please include the results of the RNA sample quality test in the methods or supplemental section to prove the reliability of the data.　For example, nanodrop and bioanalyzer tests.

3. Please add a conclusions section with your results included.

4. Please revise the discussion section to clearly show the relevance of your results.

Currently, it is difficult to understand the relationship between your results and references to previous studies.

5. You note the “unique economic value of Taihe BB-sfc chickens in the future”

Please explain in more detail. Why and how is it useful?

7. PLOS authors have the option to publish the peer review history of their article (what does this mean?). If published, this will include your full peer review and any attached files.

Reviewer #2: No

Reviewer #3: **Yes: **Samuel Nahashon

Reviewer #4: No

---

## [Author Response · Author response to Decision Letter 1]

4 Feb 2024

Reviewer #2: The authors have put together an interesting data on the muscle of Chinese Taihe Black-bone Silky Fowl.

Minor comment:

1.Line140-141: |fold changes| or |log2(fold change)| ?

Answer: Thank you for your nice reminding. We have revised it to |log2(fold change)|.

Reviewer #3: The manuscript has been revised adequately. All the issues raised have been adequately addresses including depositing of data in the NCBI.

Answer: Thank you for your positive comments.

Reviewer #4: In your research, you performed RNA-Seq, transcriptome, and proteome analyses using Black-Bone silky fowl chicken.　You identified genes from these analyses that were associated with improving the antioxidant status of muscles.　However, I have some suggestions.

1. Please make the abstract section a little more concise.

Answer: Thank you for your good suggestions. We have reduced the abstract section from 331 words to 223 words.

2. Please include the results of the RNA sample quality test in the methods or supplemental section to prove the reliability of the data.　For example, nanodrop and bioanalyzer tests.

Answer: Thank you for your nice suggestion. We have added the RNA sample quality in the supplemental Table S4.

3. Please add a conclusions section with your results included.

Answer: Thank you for your nice suggestion. We have added a conclusion section after tha discussion section.

4. Please revise the discussion section to clearly show the relevance of your results. Currently, it is difficult to understand the relationship between your results and references to previous studies.

Answer: Thank you for your nice suggestion. We have made significant modifications to the discussion section, making our research results more relevant to previous studies.

5. You note the “unique economic value of Taihe BB-sfc chickens in the future” Please explain in more detail. Why and how is it useful?

Answer: Thank you for your nice suggestion. The Chinese Taihe silky fowl chicken is a national product of geographical indication that known for its top ten characteristics, high nutritional and medicinal value. The chicken of Taihe Black-bone silky fowl is fresh and tender in meat with high protein and low fat, which the average crude protein content of 52.72% and crude fat only accounting for 24.17%. It is understood that Taihe silky fowl chicken is the main source of income for local farmers, but the main obstacle to promoting Taihe black-bone chicken is that its unique nutritional composition is not yet clear. Therefore, it is necessary to clarify the differences in muscle composition between Taihe silky fowl chicken and other black-bone chickens through transcriptome and proteome sequencing methods.

---

## [Decision Letter · Decision Letter 2]

9 Feb 2024

Integrated transcriptome and proteome analysis reveals the unique molecular features and nutritional components on the muscles in Chinese Taihe black-bone silky fowl chicken

PONE-D-23-37042R2

Dear Dr. Liao,

We’re pleased to inform you that your manuscript has been judged scientifically suitable for publication and will be formally accepted for publication once it meets all outstanding technical requirements.

Kind regards,

Tomoyoshi Komiyama, Ph.D

Academic Editor

PLOS ONE

Additional Editor Comments (optional):

Dear authors,

Thank you for submitting your revised manuscript.

I think it was much easier to understand than the original manuscript.

I am satisfied with the responses and the edits, I am happy to accept this manuscript.

The authors have replied to my remaining questions satisfactorily from four reviewers.

Therefore, I have no further comments to make, all of my previous concerns were adequately addressed.

This manuscript will be satiating the reader's interest.

Tomoyoshi Komiyama

Reviewers' comments:

Reviewer's Responses to Questions

**Comments to the Author**

1. If the authors have adequately addressed your comments raised in a previous round of review and you feel that this manuscript is now acceptable for publication, you may indicate that here to bypass the “Comments to the Author” section, enter your conflict of interest statement in the “Confidential to Editor” section, and submit your "Accept" recommendation.

Reviewer #4: All comments have been addressed

2. Is the manuscript technically sound, and do the data support the conclusions?

Reviewer #4: Yes

3. Has the statistical analysis been performed appropriately and rigorously? 

Reviewer #4: Yes

4. Have the authors made all data underlying the findings in their manuscript fully available?

Reviewer #4: Yes

5. Is the manuscript presented in an intelligible fashion and written in standard English?

Reviewer #4: Yes

6. Review Comments to the Author

Reviewer #4: Dear authors,

Thank you for submitting your revised manuscript.

The authors revised the sentences according to my comments.

Therefore, I have no further questions.

7. PLOS authors have the option to publish the peer review history of their article (what does this mean?). If published, this will include your full peer review and any attached files.

Reviewer #4: No

---

## [Editor Report · Acceptance letter]

27 Feb 2024

PONE-D-23-37042R2 

PLOS ONE

Dear Dr. Liao, 

I'm pleased to inform you that your manuscript has been deemed suitable for publication in PLOS ONE. Congratulations! Your manuscript is now being handed over to our production team.

Kind regards, 

on behalf of

Dr. Tomoyoshi Komiyama 

Academic Editor

PLOS ONE